# OpenReview forum: "Breaking Agent Backbones: Evaluating the Security of Backbone LLMs in AI Agents"
_ICLR.cc/2026/Conference — ICLR 2026 Poster_

### Official Review · Reviewer_savv · 2025-10-25

**Soundness:** 2
**Presentation:** 2
**Contribution:** 2
**Rating:** 4
**Confidence:** 3

**Summary:**

This paper introduces threat snapshots, a framework that isolates specific states in an agent’s execution flow where LLM vulnerabilities emerge. By doing so, it enables the systematic identification and categorization of security risks that propagate from the LLM to the agent level.
We apply this framework to build $B^3$, a security benchmark derived from 79,466 unique, crowdsourced adversarial attacks.

**Strengths:**

The scope of the evaluation is extensive, covering 27 popular LLMs. The benchmark construction is also interesting, as it leverages crowdsourced adversarial attacks to ensure diverse and realistic threat coverage.

**Weaknesses:**

1. It remains unclear why isolating backbone LLMs is necessary for analyzing their influence on agent security. This represents only one aspect of LLM security, and it is not very meaningful without studying the end-to-end security of the entire agent system.

2. The benchmark has limited value for evaluating newly released LLMs, as it may not fully reflect recent advancements or evolving attack surfaces.

3. The contribution appears limited because the proposed “threat snapshots” lack substantial technical novelty, and the resulting benchmark offers only modest security insights. Furthermore, the range of applicable usage scenarios remains limited.

**Questions:**

1. Do you think security issues of backbone LLMs should be addressed at the model level or within the broader agent system? If mitigations are implemented primarily in other components of the agent, does identifying weaknesses in the model itself remain sufficiently meaningful for model developers?

2. When a new LLM is released, how should its security be evaluated within this benchmarking framework?

---

> ### Author Response · Authors · 2025-11-21
>
> Thank you for your assessment of our work.
>
> We address each of the weaknesses you mentioned separately.
>
> First, regarding the usefulness of isolating the backbone LLM via TSs. Isolating specific states comes with the following benefits: (1) This approach simplifies modeling and evaluation. Despite the rapid proliferation of diverse agentic frameworks and implementations, we only need to model a single LLM call instead of simulating a full agent in these different frameworks. For the b³-benchmark, this enables broader coverage across attack categories and more efficient benchmarking and attack collection compared to running full agents. Evaluating security in a single agentic framework is narrow and may provide insights that do not generalize to other frameworks; a TS-based evaluation does not suffer from this. (2) By isolating states in an agent, we gain visibility on the specific vulnerability and can address them more targeted. For example, we can easily compare how different processing steps (e.g., on user messages or external data) affect security. (3) As we discuss in the "Response to all reviewers" even though we only consider individual TSs, we can still use this to lower bound the compound risk of the e2e system.
>
> Second, regarding evaluating newly released models. Our benchmark applies directly to newly released models. In the updated manuscript, we illustrate this by adding 9 newly released models to the evaluation. Importantly, we specifically selected the attacks to be generalizable (in particular, we now updated the attack selection to stratify across sessions to ensure more diverse attacks that should provide better generalizability). Moreover, while the current version of the benchmark fixes attacks and TSs, it can be easily extended along both dimensions to capture new threats and include more challenging (or even automatically generated) attacks.
>
> Third, regarding novelty of the threat snapshot framework. We disagree with the characterization that the TS framework lacks substantial technical novelty. Using the statelessness of AI agents to simplify evaluating their security is novel and has – to the best of our knowledge – not been considered before. By providing a benchmark that directly compares the contextual security properties of models provides observability of how and whether models are improving in terms of security and allows developers to select backbone models that are most secure for their application.
>
>
> Below we respond to your questions:
> > Question 1: Do you think security issues of backbone LLMs should be addressed at the model level or within the broader agent system? If mitigations are implemented primarily in other components of the agent, does identifying weaknesses in the model itself remain sufficiently meaningful for model developers?
>
> We believe there might have been two small misunderstanding (but please correct us otherwise):
> 1. TSs capture LLM vulnerabilities of the backbone LLM ‘within’ a full agentic system. Any external processing steps that occur in the agent are captured via the model context and the attack insertion method in the TS. For example, if there is an input sanitization that limits user inputs to 100 characters, this can (and should!) be captured by the TS via the attack insertion.
> 2. Our benchmark applies threat snapshots to evaluate how vulnerable different LLMs are across a comprehensive range of TSs that may appear in real-world applications. For this purpose, we attempt to construct TSs with diverse coverage and compare how well each model would protect against the specific threats outlined in the objectives (given that everything else remains equal).
> So, to answer your questions directly: We believe security issues should be addressed at the agent system level, and it is meaningful to determine the LLM weaknesses (in the backbone model) while it is embedded ‘within’ a specific agentic system. Threat snapshots allow us to do exactly that.
>
> > Question 2: When a new LLM is released, how should its security be evaluated within this benchmarking framework?
>
> Please see our response to the second weakness above.

---

> > ### Comment · Reviewer_savv · 2025-11-24
> >
> > Thank you for your response, which was very helpful. However, I'm still quite confused. Let's take content injection as an example. There are existing benchmarks designed to test prompt injection attacks on LLMs. Can benchmarks for LLM prompt injection adequately represent the security risks when LLMs are deployed in agent systems? My concern is that even if a prompt injection vulnerability goes undetected in an agent context, it could still be exploited in a future agent design. This raises the question: why is agent-specific benchmarking considered necessary if standard LLM benchmarks are provided?
> >
> > Thanks again for your response.

---

> > > ### Author Response · Authors · 2025-11-25
> > >
> > > You raise an important point about the relationship between agent-specific and LLM prompt-injection benchmarks.
> > >
> > > A crucial distinction is given in terms of scope. In our work, AI agents encompass essentially any software that embeds LLMs within its execution flow. Critically, this means agents are a superset of the simpler LLM applications tested in existing prompt injection benchmarks (e.g., RAG applications or chatbots). Because agents encompass more than simple LLM applications, they expose attack surfaces that standard LLM benchmarks are not designed to evaluate.
> > >
> > > However, you correctly identify a critical limitation of agent benchmarks that use full agent simulations: vulnerabilities may go undetected in one agent design but remain exploitable in future designs. This is precisely why we advocate for evaluating LLM vulnerabilities in isolation via TSs; it provides a more future-proof and generalizable security evaluation.
> > >
> > > Despite reducing to single-step LLM evaluations, b^3 is fundamentally an agentic benchmark. Agent systems expose a distinct threat surface that doesn't exist in simple LLM applications (e.g., vulnerabilities coming from tool use or structured outputs). Existing LLM benchmarks that ignore this agentic context provide only partial coverage of what matters in real-world agent deployments.
> > >
> > > The TS framework systematically identifies these agent-specific threats by considering the full agentic context (see our attack categorization), then distills each vulnerability into a focused single-step test case. This design gives us the best of both worlds: comprehensive coverage of agent-specific vulnerabilities while maintaining generalizability across different agent implementations.
> > >
> > > To directly address your questions:
> > > > Can benchmarks for LLM prompt injection adequately represent the security risks when LLMs are deployed in agent systems?
> > >
> > > Yes, but only if they provide coverage across the full threat surface. We argue that our b^3 benchmark (via the attack categorization) achieves this comprehensive coverage, whereas existing prompt injection benchmarks fall short.
> > >
> > > > Why is agent-specific benchmarking considered necessary if standard LLM benchmarks are provided?
> > >
> > > The agent perspective is necessary because it allows us to systematically identify and encompass all LLM-based threats that can occur in any software embedding LLMs.

---

> > > > ### Comment · Reviewer_savv · 2025-11-27
> > > >
> > > > Thanks for your response. Now I agree that this paper offers valuable contributions and insights, and I am fine with its acceptance.
> > > >
> > > > However, I would still like to raise one concern: I do not recommend claiming that the coverage is “comprehensive.” There is substantial prior work that exploits backbone LLM vulnerabilities—revealed by standard prompt-injection benchmarks or other threat benchmarks—to develop new attacks on backbone-LLM-based agents. This indicates that (1) many existing agent attacks are essentially derived from LLM-level vulnerabilities revealed by LLM-specific benchmarks, and (2) vulnerabilities exposed solely through agent-level evaluation may not be fully comprehensive, since future work could further exploit backbone LLM vulnerabilities identified by standard LLM benchmarks. This is why I previously mentioned “future agent design.” It concerns not only the design of agent architectures but also the design of agent-level attacks that better or further exploit existing vulnerabilities in LLMs.
> > > >
> > > > My reasoning for this “incompleteness” follows a common idea-development pattern: (1) identify vulnerabilities in backbone LLMs through standard benchmarks, and then (2) exploit those vulnerabilities in current agent designs or archs.
> > > >
> > > > That said, I fully agree that an agent-level benchmark can surface vulnerabilities in backbone LLMs that standard LLM benchmarks fail to reveal. This is an important advantage, and it makes the benchmark more meaningful.
> > > >
> > > > In sum, the key question is: for vulnerabilities rooted in backbone LLMs, what additional aspects are exposed by an agent-level benchmark compared to a standard LLM benchmark? The answer does not need to be exact, but a discussion would be helpful for future readers. Could the authors address this in the manuscript?

---

> > > > > ### Author Response · Authors · 2025-11-27
> > > > >
> > > > > Thanks. We have now made two additional changes (in green) to the manuscript to address your concerns:
> > > > > 1) In Section 2.2.2 (first paragraph) we now added a sentence that highlights the need for an agent perspective and lists the additional aspects that are exposed by an agent-level benchmark.
> > > > > 2) In Section 3.1 we now added a sentence explaining that there might be blind spots in the benchmark even though we aim to be comprehensive.

---

### Official Review · Reviewer_AxvH · 2025-10-28

**Soundness:** 3
**Presentation:** 3
**Contribution:** 3
**Rating:** 6
**Confidence:** 4

**Summary:**

This paper introduces threat snapshots, a framework for systematically evaluating the security of backbone LLMs in AI agents. Instead of modeling full agent workflows, it analyzes key states where attackers can inject malicious context. Using the b³ benchmark—210 curated attacks across 10 scenarios—the authors test 27 LLMs and find that reasoning modes often improve robustness, while larger models are not necessarily more secure. The study highlights key vulnerabilities, proposes standardized evaluation metrics, and calls for security to be treated as a core dimension of LLM assessment.

**Strengths:**

1. Proposes a novel threat snapshot framework to systematically analyze LLM security in agents.

1. Builds the b³ benchmark with diverse, realistic attack scenarios and defense levels.

1. Uses crowdsourced red-teaming to collect over 79,000 real attack prompts.

1. Reveals that reasoning modes improve robustness, while size doesn’t guarantee safety.

1. Offers clear practical insights for designing safer AI agents.

**Weaknesses:**

1. The paper does not evaluate model utility or performance trade-offs, making it hard to identify backbones that balance security and capability.

1. The tested defenses are limited, excluding some sota defense methods.

1. The threat snapshot abstraction only captures single-step interactions, overlooking multi-turn or long-horizon attacks that occur in real agents.

**Questions:**

1. There are already several benchmarks for evaluating agent security. How does your b³ benchmark differ from existing ones in scope, methodology, or attack coverage? A comparison table would help clarify the distinctions.

2. The selection of threat snapshots in Table 2 seems heuristic. How can you ensure that these categories are comprehensive enough to capture new or unseen agent scenarios in the future?

3. Could you evaluate or at least discuss more state-of-the-art defense methods beyond the three levels (L1–L3)? What insights from your attack results could inform the design of stronger defenses?

4. How were the tools and their functionalities implemented in your benchmark? Were they adapted from existing agent frameworks or custom-designed for this study?

---

> ### Author Response · Authors · 2025-11-21
>
> Thank you for your assessment of our work.
>
> You identified three weaknesses: (1) utility-analysis, (2) missing defenses, and (3) multi-agent/step settings. We address them as follows: (1) Most LLM benchmarks today focus on utility; we intentionally focus on security to fill this gap. We agree that considering the security-utility tradeoff is crucial, so we now added Figure 13 (in Appendix D.3 and referenced in Sections 4.2 and 5), which plots vulnerability scores against the agent intelligence index for different models. We have now added a few words on this topic to the Discussion.  (2) We only include LLM-based defenses (e.g., LLM self-judges) that directly depend on the backbone. Adding external SOTA defenses would conflate backbone security with defense mechanisms, moving away from our goal of isolating backbone vulnerabilities for model selection. That said, as we mention at the end of Section 2.2.1, practitioners can also use the TS framework to evaluate/red-team different defenses in a fixed AI agent – this is, however, beyond the scope of this paper. (3) Please see our general response on "Response to all reviewers".
>
> Below we respond to your questions:
> > Question 1: There are already several benchmarks for evaluating agent security. How does your b³ benchmark differ from existing ones in scope, methodology, or attack coverage? A comparison table would help clarify the distinctions.
>
> We agree that a more detailed comparison would strengthen the manuscript and have now added one in Appendix H and reference it when discussing related works in Section 1. In brief the main differences are: (1) b^3 considers threat snapshots instead of full agent simulations, hence more clearly separates LLM vulnerabilities from traditional software vulnerabilities, does not rely on a specific agentic framework, and provides sub-rankings along meaningful attack categories. (2) b^3 is based on strong human-generated adversarial attacks targeted to a specific attack objective, rather than templated attacks, making the benchmark more adversarial, which we believe is a must have for an accurate assessment of system security. (3) Broad coverage of our security-related attack categorization.
>
> > Question 2: The selection of threat snapshots in Table 2 seems heuristic. How can you ensure that these categories are comprehensive enough to capture new or unseen agent scenarios in the future?
>
> We ensured that the selected TSs were comprehensive by first designing two attack categorizations (see Appendix A), then designing the TSs to cover all categories. We carefully validated the completeness of our attack categorization and welcome feedback on potential gaps.
> Additionally, we empirically evaluated how well the TSs generalize in Appendix C.2: For this we constructed 10 additional TSs with modified app descriptions and attack objectives and then adapted our attacks (using an LLM) to the new objectives. As shown in Figure 4, the resulting ranking was relatively stable and appears to be more affected by the quality of the attacks rather than the changes in the threat snapshots.
>
> > Question 3: Could you evaluate or at least discuss more state-of-the-art defense methods beyond the three levels (L1–L3)? What insights from your attack results could inform the design of stronger defenses?
>
> We agree with the reviewer that considering additional defenses is interesting. The TS framework can naturally incorporate any defense mechanisms, including external ones. However, our specific application (the b³-benchmark) aims to isolate vulnerabilities in the backbone LLM itself, so we intentionally exclude external defenses to avoid conflating different security aspects. L1-L3 are specifically designed as model-internal defenses (e.g., we consider only an LLM 'self'-judge) to maintain this focus on backbone quality.
>
> > Question 4: How were the tools and their functionalities implemented in your benchmark? Were they adapted from existing agent frameworks or custom-designed for this study?
>
> One of the advantages of the TS framework is that it is sufficient to provide only the tool descriptions (here we use the JSON format used in MCPs) without needing to implement specific tools – the LLM only decides whether and how to call a tool. In threat snapshots where the LLM consumes tool outputs, we mock the outputs realistically. We designed all tools ourselves to match each of the applications, while taking inspiration from existing tools.

---

### Official Review · Reviewer_u1Ga · 2025-10-30

**Soundness:** 3
**Presentation:** 3
**Contribution:** 3
**Rating:** 8
**Confidence:** 3

**Summary:**

This paper proposes a new framework, threat snapshots, for evaluating the security of Large Language Models (LLMs) used as backbones in AI agents. The framework isolates specific states in an agent's execution flow, decoupling the LLM's vulnerabilities from traditional software risks. Based on this framework, the authors construct $b^3$, a large-scale security benchmark. This benchmark includes 10 threat scenarios and an impressive dataset of 79,466 adversarial attacks collected via a gamified crowdsourcing challenge. The authors then use $b^3$ to conduct a comprehensive evaluation of 27 popular LLMs. The results yield several insights, notably that enhanced reasoning capabilities, rather than model size, correlate positively with better security.

**Strengths:**

1. **Novel and Practical Framework**: The threat snapshot concept is a clear and effective abstraction. It intelligently decomposes the complex problem of "agent security" into a more manageable one: evaluating the backbone LLM's security at a specific, contextualized state. This approach greatly simplifies the evaluation process.

2. **Massive, High-Quality Benchmark ($b^3$)**: The core contribution is the $b^3$ benchmark. The dataset of nearly 80,000 human-generated adversarial attacks, collected through a "gamified" crowdsourcing effort, is a significant contribution.

3. **Broad Evaluation and Interesting Findings**: The evaluation across 27 LLMs is comprehensive. The findings provide valuable, quantifiable evidence for agent developers. While some findings confirm existing intuitions (e.g., closed-source models perform better), the conclusion that model size does not correlate with security is an important insight for the community.

**Weaknesses:**

1. **Scope Limited to Single-Agent Scenarios**: While the paper mentions that the framework could apply to Multi-Agent Systems (MAS), the 10 threat snapshots (Table 2) all focus exclusively on single-agent contexts. The evaluation misses key MAS-specific security risks, such as inter-agent deception, manipulation, or collusion.
2. **Inability to Capture Long-Horizon Attacks**: The "threat snapshot" method, by design, evaluates security at a single point in time. This makes it difficult to capture multi-step attacks, where a vulnerability only manifests after a long sequence of seemingly benign interactions. The paper claims that multi-step attacks can be "decomposed" into snapshots, but this is not experimentally validated and seems insufficient for attacks that rely on stateful, long-term manipulation.

**Questions:**

1. **Suggestion for Multi-Agent Systems (MAS)**: Given the importance of MAS, I suggest the authors discuss the challenges of applying the "threat snapshot" framework to multi-agent scenarios. For future work, adding snapshots that model inter-agent communication (e.g., one agent attempting to deceive another) would significantly broaden the benchmark's impact.

2. **Suggestion for Long-Horizon Attacks**: How does the "snapshot" framework propose to handle attacks that are stateful and build up over many turns? Would a new methodology be required to measure an agent's security "drift" during a continuous, multi-step interaction, rather than just at a single point? A discussion on this limitation would be welcome.

---

> ### Author Response · Authors · 2025-11-21
>
> Thank you for your assessment of our work.
>
> Regarding the weaknesses you mention: Please see our general response in "Response to all reviewers" which directly addresses both points. In short: (1) While our current benchmark focuses on single-backbone agents, the TS framework itself can decompose MAS attacks stepwise. (2) We acknowledge that long-horizon attacks are not directly evaluated, but single-step assessments provide a lower bound on multi-step risk and identify vulnerable components for targeted improvement. We have now clarified these limitations and the framework's extensibility in the revision.
>
> Below we respond to your questions/suggestions:
> > Question 1: Suggestion for Multi-Agent Systems (MAS): Given the importance of MAS, I suggest the authors discuss the challenges of applying the "threat snapshot" framework to multi-agent scenarios. For future work, adding snapshots that model inter-agent communication (e.g., one agent attempting to deceive another) would significantly broaden the benchmark's impact.
>
> Thank you for the suggestion. We have now added an explicit example of how to use TSs in a MAS setting in Appendix G.
>
> > Question 2: Suggestion for Long-Horizon Attacks: How does the "snapshot" framework propose to handle attacks that are stateful and build up over many turns? Would a new methodology be required to measure an agent's security "drift" during a continuous, multi-step interaction, rather than just at a single point? A discussion on this limitation would be welcome.
>
> As mentioned above, TSs can be used to model multi-step settings as well, and our benchmark can be seen as providing a lower bound on multi-step risk. To make this more explicit we have now added an example of how to use TSs in multi-step settings in Appendix B.2.

---

> > ### Comment · Reviewer_u1Ga · 2025-11-27
> >
> > Thank you for your rebuttal. My concerns have been addressed. I noticed that Appendix G focuses on comparing other agent security benchmarks, whereas the 'explicit example of how to use TSs in a MAS setting' seems to be located in Appendix B.2 (line 916).

---

> > > ### Author Response · Authors · 2025-11-27
> > >
> > > You are correct, we flipped the references in our response. Thanks.

---

### Official Review · Reviewer_gAuZ · 2025-11-01

**Soundness:** 3
**Presentation:** 3
**Contribution:** 3
**Rating:** 6
**Confidence:** 3

**Summary:**

This paper introduces Threat Snapshots, a formal framework for systematically isolating and modeling LLM-specific vulnerabilities in AI agents, distinguishing them from traditional software security flaws. Using this abstraction, the authors construct a large-scale, open benchmark based on 79K human-crafted adversarial attacks across ten representative agentic scenarios.

**Strengths:**

1. The Threat Snapshot formalism is a major conceptual contribution that elegantly decouples LLM-specific vulnerabilities from agentic system context, enabling generalizable benchmarking and red teaming.
2. The benchmark (79K attacks, 27 models) is very comprehensive. It is valuable for both researchers and practitioners.

**Weaknesses:**

The threat-snapshot abstraction focuses on isolated LLM calls and single-backbone behavior, but the paper does not convincingly show that these snapshots still isolate backbone vulnerabilities when execution flows interleave multiple LLMs or long multi-turn interactions. In real agent deployments, control flow, state handoffs, and interaction between multiple models can create emergent attack surfaces (e.g., prompt-infection cascading across agents) that a single-call snapshot may miss. This leaves an open question about how well b³ predicts security in richer, multi-actor deployments.

**Questions:**

1. How transferable are the threat snapshot vulnerabilities to multi-turn or multi-agent settings—do they still isolate backbone behavior cleanly when control flow interleaves multiple LLMs?
2. Could the framework incorporate automated attack generation (e.g., adaptive red teaming or LLM-driven mutation) to reduce human bias in the crowdsourced dataset?

---

> ### Author Response · Authors · 2025-11-21
>
> Thank you for your assessment of our work.
>
> Regarding the weakness you mention: We address the generalizability of TSs to multi-agent deployments in detail in "Response to all reviewers" above. In brief: TSs decompose complex attack pathways (including prompt-injection cascades) into individual steps, with single-step vulnerabilities establishing a lower bound on multi-step risk. Importantly, they explicitly capture agent specific processing steps (e.g., control flows, state hand-offs etc) via the model context and attack insertion. We believe the updated manuscript makes these points clearer now.
>
> Below we respond to your questions:
> > Questions 1: How transferable are the threat snapshot vulnerabilities to multi-turn or multi-agent settings—do they still isolate backbone behavior cleanly when control flow interleaves multiple LLMs?
>
> Yes, as discussed above, TS vulnerabilities provide a lower bound on risk on the agent level security.
>
> > Question 2: Could the framework incorporate automated attack generation (e.g., adaptive red teaming or LLM-driven mutation) to reduce human bias in the crowdsourced dataset?
>
> Using automatically generated attacks is a long-term goal for this benchmark. While automatic red-teaming methods aren't yet powerful enough to reliably break these systems, our TS framework integrates with reinforcement learning frameworks, that will enable attacker models that can construct adaptive attacks that improve as backbone models advance. We've added a comment at the end of Section 5.

---

### Author Response · Authors · 2025-11-21
**Response to all reviewers**

We would like to thank all reviewers for taking the time to review our manuscript, which has helped improve it. We have now posted an updated version and, for reviewer’s convenience, highlighted all changes (except small fixes like typos and removals) in blue in the updated manuscript.

Before responding to the questions from each reviewer individually, we jointly address the concern raised by several reviewers on how the threat snapshot (TS) framework deals with multi-step attacks and multi-agent systems:

**TS framework applies to multi-turn, multi-agent attacks.** The TS framework is outcome-driven, beginning with concerning system outcomes and evaluating whether strong adversarial attacks can achieve them, regardless of the specific attack techniques used. This approach enables modeling of multi-turn and multi-agent threats by starting with a target outcome and decomposing it into a chain of threat snapshots that represent the required steps. This decomposition is possible because LLMs are stateless -- each call receives the full context needed for inference -- making TSs a complete atomic abstraction regardless of complexity. Multi-step attacks are modeled by chaining individual threat snapshots: for $\text{LLM}_1\rightarrow S_1 \rightarrow \text{LLM}_2 \rightarrow S_2 \rightarrow \text{objective}$, we identify what output from $\text{LLM}_2$ compromises the attack objective, then what output from $\text{LLM}_1$ produces that outcome (where $S_i$ represent the processing steps that transform outputs into new input). In Appendix B.2, we provide concrete examples of such decompositions for multi-agent and multi-turn settings. We have added corresponding clarifications in Section 2.2.1 (lines 207-215).

**The $b^3$ benchmark focuses on single-step threat snapshots by design.** Our core problem is LLM backbone selection: how can developers choose the most secure backbone for their agentic system? Single-step evaluation directly addresses this need. Comprehensive coverage of attack categories is essential for meaningful backbone comparison, whereas multi-step chaining creates exponential complexity that forces other benchmarks to sacrifice breadth. We prioritize holistic single-step coverage over sparse multi-step scenarios, believing this provides more actionable information for backbone selection. Moreover, we expect that models weaker in single-step settings will be weaker in multi-step scenarios, so $b^3$ establishes a meaningful risk lower bound even for complex agentic systems. Finally, our approach enables developers to identify vulnerable components and create sub-rankings based on specific capabilities (tool calling, indirect attacks, etc.), allowing them to select backbones optimized for their particular use case rather than relying solely on aggregate scores. We have added corresponding sentences in Section 3.1 (lines 292-295).

---

### Meta-Review · Area_Chair_7xcD · 2026-01-08

**Summary:**

This paper presents a Threat Snapshots (TS) framework for evaluating the security of backbone LLMs in AI agents by isolating specific vulnerability states rather than simulating full agent workflows. With this framework, the authors construct $b^3$, a security benchmark based on 194K crowdsourced adversarial attacks, and evaluate 34 (initially 27) popular LLMs with it. The reviewers praised the work's novelty and contribution, the quality of the benchmark, and the comprehensiveness of the evaluation. They also raised some concerns on the validity of single-step snapshots for evaluating complex agentic systems and potential limitations on multi-agent and long-horizon tasks.

**Reviewer Concerns:**

The authors provided detailed and convincing rebuttal to reviewer concerns. There are no major outstanding concerns.

**Reviewer Scores:**

Reviewer u1Ga has an initial score of 8 and responded to the rebuttal saying that their concerns have been addressed. Reviewer savv has an initial score of 4. They responded to the rebuttal and stated that they would be "fine with its acceptance" after a couple of exchanges with the authors. Reviewers gAuZ and AxvH have positive initial scores (6) and did not respond. They would likely maintain or increase their scores given the quality of the rebuttal.

---

### Decision · Program_Chairs · 2026-01-26

Accept (Poster)